# Tunnel Try-on: Excavating Spatial-temporal Tunnels for High-quality Virtual Try-on in Videos

**Zhengze Xu***
School of AIA, Huazhong University
of Science and Technology
Wuhan, China
zhengzexu@hust.edu.cn

**Mengting Chen†**
Alibaba
Hangzhou, China
cmt271286@alibaba-inc.com

**Zhao Wang**
Alibaba
Hangzhou, China
mingzhao.wz@alibaba-inc.com

**Linyu Xing**
Alibaba
Hangzhou, China
xinglinyu.xly@alibaba-inc.com

**Zhonghua Zhai**
Alibaba
Hangzhou, China
zhaizhonghua.zzh@alibaba-inc.com

**Nong Sang**
State Key Lab of MIIPT, Huazhong
University of Science and Technology
Wuhan, China
nsang@hust.edu.cn

**Jinsong Lan**
Alibaba
Beijing, China
jinsonglan.ljs@alibaba-inc.com

**Shuai Xiao‡**
Alibaba
Hangzhou, China
shuai.xsh@alibaba-inc.com

**Changxin Gao‡**
State Key Lab of MIIPT, Huazhong
University of Science and Technology
Wuhan, China
cgao@hust.edu.cn

## Abstract

Video try-on is challenging and has not been well tackled in previous works. The main obstacle lies in preserving the clothing details and modeling the coherent motions simultaneously. Faced with those difficulties, we address video try-on by proposing a diffusion-based framework named "Tunnel Try-on." The core idea is excavating a "focus tunnel" in the input video that gives close-up shots around the clothing regions. We zoom in on the region in the tunnel to better preserve the fine details of the clothing. To generate coherent motions, we leverage the Kalman filter to smooth the tunnel and inject its position embedding into attention layers to improve the continuity of the generated videos. In addition, we develop an environment encoder to extract the context information outside the tunnels. Equipped with these techniques, Tunnel Try-on keeps fine clothing details and synthesizes stable and smooth videos. Demonstrating significant advancements, Tunnel Try-on could be regarded as the first attempt toward the commercial-level application of virtual try-on in videos. The Project page is here.

## CCS Concepts

• **Computing methodologies** → **Computer vision**.

*Work done during an internship at Alibaba.
†Project leader.
‡Corresponding authors.

## Keywords

Virtual Try-On, Diffusion Models, Video Synthesis

**ACM Reference Format:**
Zhengze Xu, Mengting Chen, Zhao Wang, Linyu Xing, Zhonghua Zhai, Nong Sang, Jinsong Lan, Shuai Xiao, and Changxin Gao. 2024. Tunnel Try-on: Excavating Spatial-temporal Tunnels for High-quality Virtual Try-on in Videos. In *Proceedings of the 32nd ACM International Conference on Multimedia (MM '24), October 28-November 1, 2024, Melbourne, VIC, Australia.* ACM, New York, NY, USA, 10 pages. https://doi.org/10.1145/3664647.3680836

## 1 Introduction

Video virtual try-on aims to dress the given clothing on the target person in videos. It requires to preserve both the appearance of the clothing and the motions of the person. It provides an interactive experience, enabling consumers to explore clothing options without the necessity for physical try-on, which has garnered widespread attention from both the fashion industry and consumers alike.

Although there are not many studies on video try-on, image-based try-on have already been extensively researched. Numerous classical image virtual try-on methods rely on the Generative-Adversarial-Networks(GANs) [7, 9, 10, 20, 39]. These methods typically comprise two primary components: a warping module that warps clothing to fit the person in semantic level and a try-on generator that blends the warped clothing with the person image. Recently, diffusion models [33] have significantly improved the quality of image and video generation. Some diffusion-based methods [23, 51] for image virtual try-on have been proposed, which do not explicitly incorporate a warp module but instead integrate the warp and blend process in a single unified process. Leveraging pre-trained text-to-image diffusion models, these diffusion-based models achieve fidelity surpassing that of GAN-based models.

Video try-on provides a more comprehensive presentation of the try-on clothing under different conditions compared to image

Input Video      Clothing                      Synthetic Video

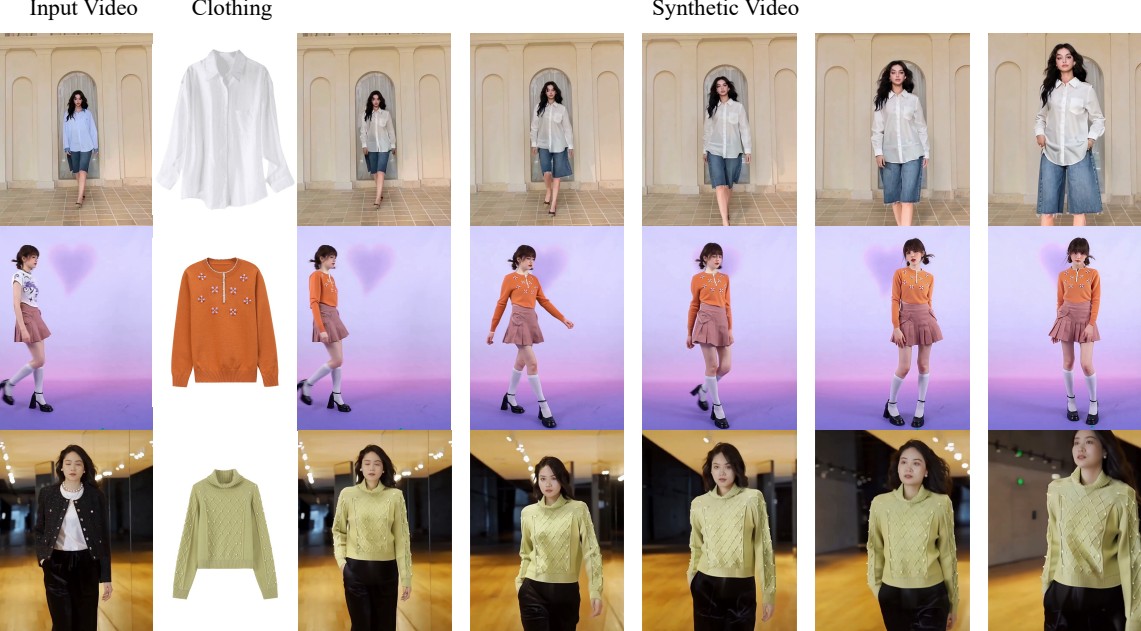

**Figure 1: Results of Tunnel Try-on. It achieves state-of-the-art performance, which can not only handle complex clothing and backgrounds but also adapt to different types of movements (first and second rows) and camera angle changes (third row).**

try-on. A direct transfer approach is to apply image try-on methods to process videos frame by frame. However, this leads to significant inter-frame inconsistency, resulting in unacceptable generation outcomes. Several approaches have explored specialized designs for video virtual try-on [8, 21, 25, 50]. These methods typically utilize optical flow prediction modules to warp frames generated by the try-on generator, aiming to enhance temporal consistency. Cloth-Former [21] additionally proposes temporal smoothing operations for the input to the warping module. While these explorations make steady advancements, most of them only tackle simple scenarios. For example, in VVT [8] dataset, samples mainly include simple textures, tight-fitting T-shirts, plain backgrounds, fixed camera angles, and repetitive human movements. This notably lags behind image virtual try-on and falls short of meeting practical application needs. We analyze that, different from the image-based settings, the main challenge in video try-on is preserving the fine detail of the clothing and generating coherent motions at the same time.

In this paper, to address the aforementioned challenges in complex natural scenes, we propose a novel framework termed Tunnel Try-on. We start with a strong baseline of image-based virtual try-on. It leverages an inpainting U-Net (noted as Main U-Net) as the main branch and utilizes a reference U-Net (noted as Ref U-Net) to extract and inject the fine details of the given clothing. By inserting motion modules after each stage of the Main U-Net, we extend this model to conduct video try-on. However, this basic solution is insufficient to deal with the challenging cases in real-world videos.

We observe that the human often occupies a small area in videos and the area or location could change violently along with the camera movements. Thus, we propose to excavate a "tunnel" in the given video to provide a stable close-up shot of the clothing region. Specifically, we conduct a region crop in each frame and zoom in

on the cropped region to ensure that the individuals are appropriately centered. This strategy maximizes the model's capabilities for preserving the fine details of the reference clothing. At the same time, we leverage Kalman filtering [42] to calculate the coordinates of the cropping boxes and inject the position embedding of the focus tunnel into the motion module. In this way, we could keep the smoothness and continuity of the cropped video region and assist in generating more consistent motions. Additionally, although the regions inside the tunnel deserve more attention, the outside region could provide the global context for the background around the clothing. Thus, we develop an environment encoder. It extracts global high-level features outside the tunnels and incorporates them into the Main U-Net to enhance the background generation.

Extensive experiments demonstrate that equipped with the aforementioned techniques, our proposed Tunnel Try-on significantly outperforms other video virtual try-on methods. In summary, our contributions can be summarized in the following three aspects:

- We proposed Tunnel Try-on, the first diffusion-based video virtual try-on model achieving state-of-the-art performance.
- We design a novel technique called focus tunnel to emphasize the clothing region and generate coherent motion in videos.
- We further develop several enhancing strategies like incorporating the Kalman filter to smooth the focus tunnel, leveraging the tunnel position embedding and environment context in the attentions to improve the generation quality.

## 2 Related Work

### 2.1 Image virtual try-on

Image virtual try-on methods can generally be divided into two categories: GAN-based methods [7, 10, 13, 20, 26, 29, 36, 39] and

diffusion-based methods [1, 4, 11, 23, 28, 51]. GAN-based methods typically utilize Conditional Generative Adversarial Network (cGAN) [27] and have two decoupled modules: a warping module to fit the clothing to human body and a GAN-based generator to blend the clothing with the body. They achieve accurate wrapping by estimating dense flow maps or using alignment strategies. VITON [13] proposes a coarse-to-fine strategy to warp the clothing onto the target region. CP-VTON [39] preserves the clothing identity with a geometric matching module. PBAFN [10] employs knowledge distillation for a parser-free method, reducing the need for accurate masks. VITON-HD [7] adopts alignment-aware segment normalization to address misalignment issues. However, these approaches face challenges in dealing with images of persons in complex poses and intricate backgrounds. Moreover, cGANs struggle with significant spatial transformations between the clothing and the person's posture. The exceptional generative capabilities of diffusion have inspired several diffusion-based image try-on methods. TryOnDiffusion [51] employs a dual U-Nets architecture for image try-on, which requires extensive datasets for training. Subsequent methods leverage large-scale pre-trained diffusion models as priors in the try-on networks [16, 33, 44]. LADI-VTON [28] treats clothing as pseudo-words. DCI-VTON [11] integrates clothing into pre-trained diffusion models using warping networks. StableVITON [23] conditions the intermediate feature maps of the Main U-Net with a zero cross-attention block. These methods achieve high-fidelity single-image inference, but when applied to video virtual try-on, the lack of inter-frame relationship consideration leads to significant inconsistency, resulting in unacceptable generation results.

## 2.2 Video virtual try-on

Compared to image-based try-on, video virtual try-on offers users more freedom and a more realistic try-on experience. However, few studies have explored this area to date. FW-GAN [8] and FashionMirror [3] use optical flow to warp past frames for coherent video, with the latter warping at the feature level instead of the pixel level. MV-TON [50] adopts a memory refinement module to remember the previously generated frames. ClothFormer [21] proposes a dual-stream transformer architecture to extract and fuse the clothing and the person's features. It uses a tracking strategy based on optical flow and ridge regression to obtain temporally consistent warps. Due to the difficulties faced by warp modules in handling complex textures and significant motion, previous video try-on methods are limited to handling simple cases like minor movements, simple backgrounds, and clothing with simple textures, focusing mainly on tight-fitting tops. These limitations make them inadequate for real-world scenarios involving diverse clothing types, complex backgrounds, free-form movements, and variations in the size, proportion, and position of individuals. Therefore, we propose to remove explicit warp modules and utilize diffusion models for video try-on, along with a focus tunnel strategy to adapt to varied relationships between individuals and backgrounds.

## 2.3 Image Animation

Image animation aims to generate video sequences from static images. Recently, some diffusion-based models have shown unprecedented success [6, 18, 19, 22, 30, 40, 43, 45, 47, 49]. Among

them, Magic Animate [43] and Animate Anyone [18] show the best results. Both models use an additional U-Net to extract appearance information and an encoder for pose sequences. Combining these animation frameworks with advanced image try-on methods can enable video try-on. However, this pipeline lacks guidance from actual video information, often resulting in static backgrounds that make it difficult for characters to blend into real environments. Additionally, relying solely on pose-driven actions can lead to strange generation results when conducting virtual try-on with significant changes in the magnitude or position of the person's movements.

## 3 Method

In Section 3.1, we introduce the foundational knowledge of latent diffusion models required for subsequent discussions. Section 3.2 details the network architecture of our Tunnel Try-on. In Section 3.3, we present details of the focus tunnel extraction strategy. In Section 3.4, we introduce the enhancing strategies for the focus tunnel, including tunnel smoothing and tunnel embedding. In Section 3.5, we elaborate on the environment encoder which aims at extracting the global context as the complementary. At last, we summarize our training and validation pipeline in Section 3.6.

## 3.1 Preliminaries

Diffusion models [17] have demonstrated promising capabilities in both image and video generation. Built on the Latent Diffusion Model (LDM), Stable Diffusion [33] conducts denoising in the latent space of an auto-encoder. Trained on the large-scale LAION dataset [35], Stable Diffusion demonstrates excellent generation performance. Our network is built upon Stable Diffusion.

Given an input image $\mathbf{x}_0$, the model first employs a latent encoder [24] to project it into the latent space: $\mathbf{z}_0 = \mathcal{E}(\mathbf{x}_0)$. Throughout the training, Stable Diffusion transforms the latent representation into Gaussian noise by applying a variance-preserving Markov process [37] to $\mathbf{z}_0$, which can be formulated as:

$$\mathbf{z}_t = \sqrt{\bar{\alpha}_t}\mathbf{z}_0 + \sqrt{1 - \bar{\alpha}_t}\boldsymbol{\epsilon}, \ \boldsymbol{\epsilon} \sim \mathcal{U}([0, 1]) \tag{1}$$

where $\bar{\alpha}_t$ is the cumulative product of the noise coefficient $\alpha_t$ at each step. Subsequently, the denoising process learns the prediction of noise $\boldsymbol{\epsilon}_\theta(\mathbf{z}_t, \mathbf{c}, t)$, which can be summarized as:

$$\mathbb{L}_{LDM} = \mathbb{E}_{\mathbf{z},\mathbf{c},\boldsymbol{\epsilon},t}(|\boldsymbol{\epsilon} - \boldsymbol{\epsilon}_\theta(\mathbf{z}_t, \mathbf{c}, t)|_2^2). \tag{2}$$

Here, $t$ represents the diffusion timestep, $\mathbf{c}$ denotes the conditioning text prompts from the CLIP [32], and $\boldsymbol{\epsilon}_\theta$ denotes the noise prediction neural networks like the U-Net [34]. In inference, Stable Diffusion reconstructs an image from Gaussian noise step by step, predicting the noise added at each stage. The denoised results are then fed into a latent decoder to regenerate images from the latent representations, denoted as $\hat{\mathbf{x}}_0 = \mathcal{D}(\hat{\mathbf{z}}_0)$.

## 3.2 Overall Architecture

This section provides a comprehensive illustration of the pipeline presented in Figure 2. We start with introducing the strong baseline for image try-on. Then, we extend it to videos by adding motion modules. Afterwards, we briefly describe our novel designs which will be elaborated on in the next sections.

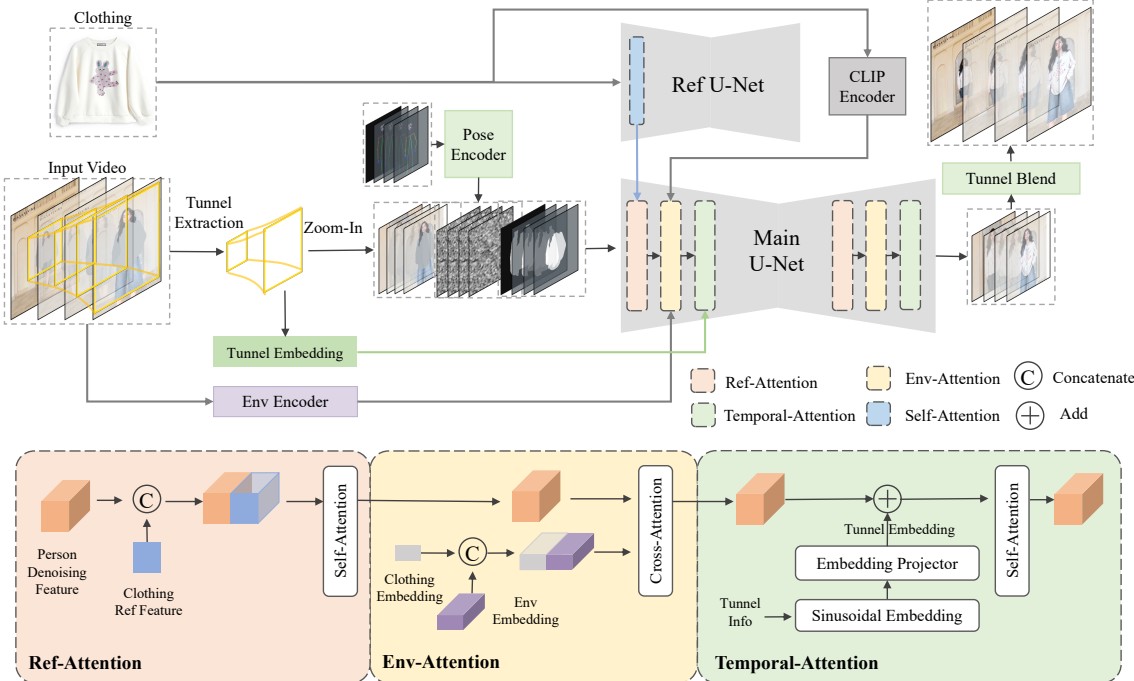

**Figure 2: The overview of Tunnel Try-on. Given an input video and a clothing image, we first extract a focus tunnel to zoom in on the garment area to better preserve the details. The zoomed region is represented by a sequence of tensors consisting of the background latent, latent noise, and the garment mask. Human pose information is added to the latent noise to assist the generation. Afterward, the 9-channel tensor is fed into the Main U-Net. A Ref U-Net and a CLIP Encoder extract the clothing image representations, which are added to the Main U-Net with Ref-attention. We also add the tunnel embedding into temporal attention for consistent motions and use an environment encoder to extract global context for additional guidance.**

*Image try-on baseline.* The baseline (modules in gray) of Tunnel Try-on consists of two U-Nets: the Main U-Net and the Ref U-Net. The Main U-Net is initialized with an inpainting model. The Ref U-Net [46] has been proven effective [4, 18, 43] in preserving detailed information of reference images. Therefore, Tunnel Try-on utilizes the Ref U-Net to encode the fine-grained features of reference clothing. Additionally, Tunnel Try-on employs a CLIP image encoder to capture high-level semantic information of target clothing images, such as overall color. Specifically, the Main U-Net takes a 9-channel tensor with the shape of $B \times 9 \times H \times W$ as input, where B, H, and W denote the batch size, height, and width. The 9 channels consist of the clothing-masked video frame (4 channels), the latent noise (4 channels), and the cloth-agnostic mask (1 channel). To enhance guidance on the movements of the generated video and further improve its fidelity, the pose maps are encoded by a pose encoder comprising several convolutions and added to the concatenated 9-channel tensor in the latent space.

*Adaption for videos.* To adapt the image try-on model for videos, we insert motion modules after each stage of the Main U-Net. Specifically, we employ Temporal-Attention, which conducts self-attention on features of the same spatial position across different frames to ensure smooth transitions between frames. The Main U-Net's feature maps are extended with the temporal dimension of $f$ (frames), changing the input shape to $B \times 9 \times f \times H \times W$. Therefore, the feature maps from the Ref U-Net are repeated $f$ times and

further concatenated along the spatial dimension after the Main U-Net. These concatenated features are then flattened along the spatial dimension and input into the self-attention module, with the output retaining only the original denoising feature map.

*Novel designs of Tunnel Try-on.* We excavate a Focus Tunnel in the input video and zoom in on the region to emphasize the clothing. To enhance the video consistency, we leverage the Kalman filter to smooth the tunnel and inject the tunnel embedding into the temporal attention layers. Simultaneously, we design an environment encoder (Env Encoder in Figure 2) to capture the global context information in each video frame as complementary cues. In this way, the Main U-Net primarily utilizes three types of attention modules to integrate control conditions at various levels, enhancing the spatio-temporal consistency of the generated video. These modules are depicted in the bottom colored box in Figure 2 and will be detailed in the following sections.

### 3.3 Focus Tunnel Extraction

In typical image virtual try-on datasets, the target person is typically centered and occupies a large portion of the image. However, in video virtual try-on, due to the movement of the person and camera panning, the person in video frames may appear at the edges or occupy a smaller portion. This can lead to a decrease in the quality of video generation results and reduce the model's ability to maintain clothing identity. To enhance the model's ability to preserve details

and better utilize the training weights learned from image try-on data, we propose the "focus tunnel" strategy, as shown Figure 2.

Specifically, depending on the type of try-on clothing, we utilize the pose map to identify the minimum bounding box for the upper or lower body. We then expand the coordinates of the obtained bounding box according to predefined rules to ensure coverage of all clothing. Since the expanded bounding box sequence resembles an information tunnel focused on the person, we refer to it as the "focus tunnel" of the input video. Next, we zoom in on the tunnel. In other words, the video frames within the focus tunnel are cropped, padded, and resized to the input resolution. Then they are combined to form a new sequence input for the Main U-Net. The generated video output from the Main U-Net is then blended with the original video using Gaussian blur to achieve natural integration.

### 3.4 Focus Tunnel Enhancement

Since the process of focus tunnel extraction is computed only within individual frames without considering inter-frame relationships, slight jitters or jumps in bounding box sequences can occur in videos due to movement of people and the camera. These jitters and jumps can result in unnatural focus tunnels compared to videos captured naturally, increasing the difficulty of temporal attention convergence and leading to decreased temporal consistency in the generated videos. Dealing with this challenge, we propose tunnel smoothing and inject tunnel embedding into the attention layers.

*Tunnel smoothing.* To smooth the focus tunnel and achieve a variation effect similar to natural camera movements, we propose the focus tunnel smoothing strategy. Specifically, we use Kalman filtering to correct the focus tunnel as Algorithm 1.

---

**Algorithm 1:** Kalman Filter.

**Input:** Raw tunnel coordinate $\mathbf{x}$, tunnel length $f$.
**Result:** Smoothed tunnel coordinate $\hat{\mathbf{x}}$.

1 Initialize $P_0 = \mathbf{x}_1, \hat{\mathbf{x}}_0 = \mathbf{x}_1, Q = 0.001, R = 0.0015, t = 1$.
2 **repeat**
3    Project the state ahead $\hat{\mathbf{x}}_t^- = \hat{\mathbf{x}}_{t-1}$.
4    Project the error covariance ahead $P_t^- = P_{t-1} + Q$.
5    Compute the Kalman Gain $K_t = P_t^- (P_t^- + R)^{-1}$
6    Update the estimate $\hat{\mathbf{x}}_t = \hat{\mathbf{x}}_t^- + K_t(\mathbf{x}_t - \hat{\mathbf{x}}_t^-)$
7    Update the error covariance $P_t = P_t^-(1 - K_t)^{-1}$
8    $t \leftarrow t + 1$.
9 **until** $t > f$;
**Output:** $\hat{\mathbf{x}}$

---

$\hat{\mathbf{x}}_t$ represents the smoothed coordinate of the focus tunnel at time $t$, calculated using the Kalman filter's prediction equation. $\mathbf{x}_t$ is the observed position of the tunnel at time $t$, i.e., the coordinate before smoothing. After the Kalman filter, we filter out the high-frequency jitter caused by exceptional cases using a low-pass filter.

*Tunnel embedding.* The input form of the focus tunnel has increased the magnitude of the camera movement. To mitigate the challenge faced by the Temporal-Attention module in smoothing out such significant camera movements, we introduce the Tunnel Embedding. Tunnel Embedding accepts a three-tuple input,

comprising the original image size, tunnel center coordinates, and tunnel size. Inspired by the design of resolution embedding in SDXL [31], Tunnel Embedding first encodes the three-tuple into 1D absolute position encoding, and then obtains the corresponding embedding through linear mapping and activation functions. Subsequently, the focus tunnel embedding is added to the temporal attention as position encoding. With Tunnel embedding, temporal attention integrates details about the size and position of the focus tunnel, aiding in preventing misalignment with focus tunnels affected by excessively large camera movements. This enhancement contributes to improving the temporal consistency of video generation within the focus tunnel.

### 3.5 Environment Feature Encoding

Applying the focus tunnel strategy can result in losing context, making it difficult to generate a reasonable background within the masked area. To address this, we propose the Environment Encoder, which consists of a frozen CLIP image encoder and a learnable linear mapping layer. The masked original image is encoded by the CLIP image encoder to capture overall environmental information, and the output features are fine-tuned through the linear projection layer. As shown in the Env-Attention of Figure 2, the output features of Environment Encoder, serving as keys and values, are injected into the denoising process through cross-attention. These features are then injected into the denoising process through cross-attention, as shown in Env-Attention in Figure 2.

### 3.6 Train and Test Pipeline

*Training process.* The training phase has two stages. In the first stage, we exclude temporal attention, the Environment Encoder, and Tunnel Embedding. We freeze the weights of the VAE encoder/decoder(omitted in Fig 2 for simplicity) and the CLIP image encoder, updating only the Main U-Net, Ref U-Net, and pose guider. In this stage, the model is trained on paired image try-on data. The objective of this stage is to learn the extraction and preservation of clothing features using larger, higher-quality, and more diverse paired image data compared to the video data, aiming to achieve high-fidelity image-level try-on generation results as a solid foundation.

In the second stage, all strategies and modules are incorporated, and the model is trained on video try-on datasets. Only the parameters of the temporal attention, Global Environment Encoder, and Focus Tunnel Position Embedding are updated in this stage. The goal of this stage is to leverage the image-level try-on capability learned in the first stage while enabling the model to learn temporally related information, resulting in high spatio-temporal consistency in try-on videos.

*Test process.* During the testing phase, the input video undergoes Tunnel Extraction to obtain the Focus Tunnel. The input video, along with the conditional videos, is then zoomed in on the focus tunnel and fed into the Main U-Net. Guided by the outputs of the Ref U-Net, CLIP Encoder, Global Environment Encoder, and Focus Tunnel Position Embedding, the Main U-Net progressively recovers the try-on video from the noise. Finally, the generated try-on video undergoes Tunnel-Blend post-processing to obtain the desired complete try-on video.

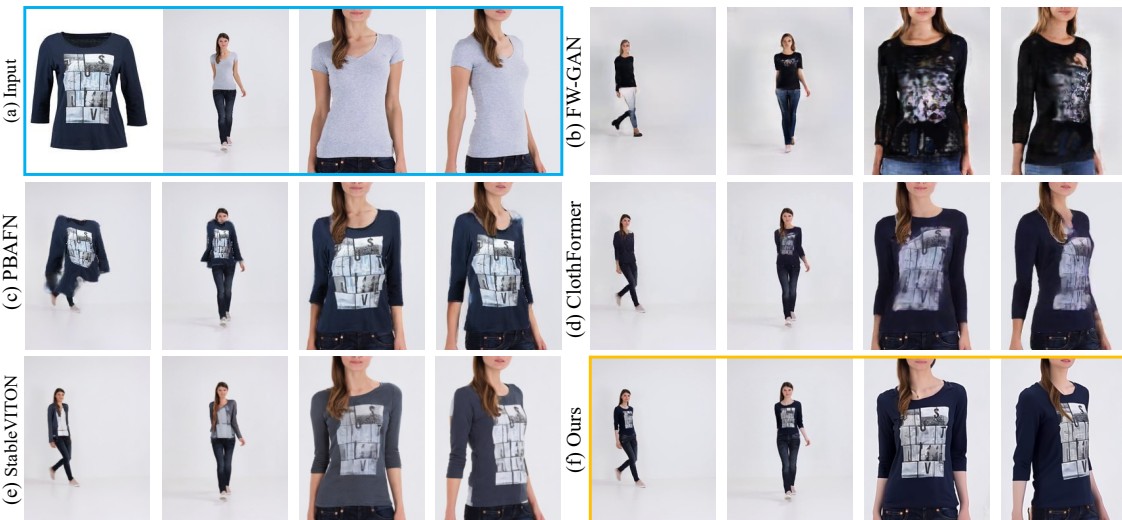

**Figure 3: Qualitative comparison with existing alternatives on the VVT dataset. The clothing and target person is shown in (a). The results of (b) FW-GAN, (c) PBAFN, (d) ClothFormer, (e) StableVITON, and (f) Tunnel Try-on are represented respectively.**

## 4 Experiments

### 4.1 Datasets

We evaluate Tunnel Try-on on two video try-on datasets: the VVT [8] dataset and our collected dataset. The VVT dataset includes 791 paired person videos and clothing images, with 192×256 resolution. The models in the videos have similar and simple poses and movements on pure white backgrounds, while the clothes are all fitted tops. Due to these limitations, it fails to reflect real-world application scenarios of virtual video try-on. Therefore, we collected a dataset from real e-commerce application scenarios, featuring complex backgrounds, diverse movements and body poses, and various types of clothing. The dataset consists of 5,350 video-image pairs. We divided it into 4,280 training videos and 1,070 testing videos, each containing 776,536 and 192,923 frames, respectively.

### 4.2 Implement Details

*Model configurations.* In our implementation, the Main U-Net is initialized with the inpainting model weight of Stable Diffusion-1.5 [33]. The Ref U-Net is initialized with a standard text-to-image SD-1.5. The Temporal-Attention is initialized from the motion module of AnimateDiff [12].

*Training and testing protocols.* The training phase is structured in two stages. In both stages, we resize and pad the inputs to a uniform resolution of 512×512, and we adopt an initial learning rate of 1e-5. The models are trained on 8x A100 GPUs. In the first stage, we utilized image try-on pairs extracted from video data, and merged them with existing image try-on datasets VITON-HD [7] for training. Then, we sample a clip consisting of 24 frames in the videos as the input for training in stage 2. In the testing phase, we use the temporal aggregation technique [38] to combine different video clips, producing a longer video output.

### 4.3 Comparisons with Existing Alternatives

We conducted a comprehensive comparison with other virtual try-on methods on the VVT dataset, including qualitative, quantitative comparisons and user studies. We collected several virtual try-on methods, covering both GAN-based methods like FW-GAN [8], PBAFN [10] and ClothFormer [21], and diffusion-based methods like Anydoor [5] and StableVITON [23]. To ensure a fair comparison, we utilized the VITON-HD [7] dataset for the first-stage training and conducted second-stage training on the VVT [8] dataset without using our own dataset.

Figure 3 displays the qualitative results of various methods on the VVT dataset. From Figure 3, it is evident that GAN-based methods like FW-GAN and PBAFN, which utilize warping modules, struggle to adapt effectively to variations in the sizes of individuals in the video. Satisfactory results are achieved only in close-up shots, with the warping of clothing producing acceptable outcomes. However, when the model moves farther away and becomes smaller, the warping module produces inaccurately wrapped clothing, resulting in unsatisfactory single-frame try-on results. ClothFormer can handle situations where the person's proportion is relatively small, but its generated results are blurry, with significant color deviation.

We also extend some diffusion-based image try-on methods (Any-Door and StableVITON) to videos by per-frame generation. We observe that they can generate relatively accurate single-frame results. However, due to the lack of consideration for temporal coherence, there are discrepancies between consecutive frames. As shown in Figure 3(e), the letters on the clothing change in different frames. Additionally, there are lots of jitters between adjacent frames in these methods, which can be observed more intuitively in videos.

Compared with those existing solutions, our Tunnel Try-on seamlessly integrates diffusion-based models and video generation models, enabling the generation of accurate single-frame try-on videos with high inter-frame consistency. As depicted in Figure 3(f), the letters on the chest of the clothing remain consistent and correct as the person moves closer.

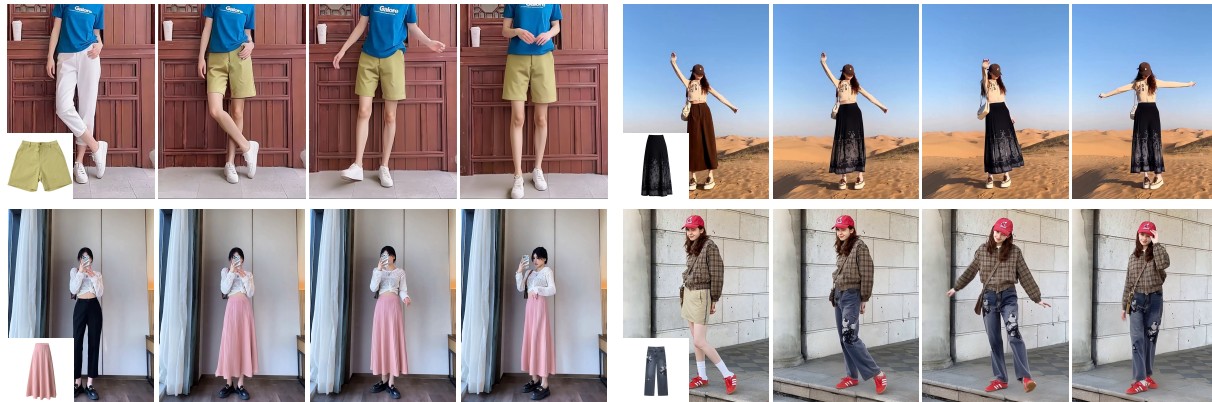

**Figure 4: Qualitative results of Tunnel Try-on on our dataset. We present the try-on results of pants and skirts, as well as cross-category try-on results.**

**Table 1: Comparison on the VVT dataset: ↑ denotes higher is better, while ↓ indicates lower is better.**

| Method | SSIM↑ | LPIPS↓ | $VFID_{I3D}$ ↓ | $VFID_{ResNeXt}$ ↓ |
|---|---|---|---|---|
| CP-VTON [39] | 0.459 | 0.535 | 6.361 | 12.10 |
| FW-GAN [8] | 0.675 | 0.283 | 8.019 | 12.15 |
| PBAFN [10] | 0.870 | 0.157 | 4.516 | 8.690 |
| ClothFormer [21] | **0.921** | 0.081 | 3.967 | 5.048 |
| AnyDoor [5] | 0.800 | 0.127 | 4.535 | 5.990 |
| StableVITON[23] | 0.876 | 0.076 | 4.021 | 5.076 |
| **Tunnel Try-on** | 0.913 | **0.054** | **3.345** | **4.614** |

**Table 2: User study for the preference rate on the VVT test dataset. \* indicates testing was conducted only on examples shown in ClothFormer demonstrations.**

| Method | Quality% | Fidelity% | Smoothness% |
|---|---|---|---|
| FW-GAN [8] | 0 | 0 | 5.62 |
| PBAFN [10] | 6.77 | 8.77 | 6.31 |
| AnyDoor [5] | 7.85 | 7.08 | 0 |
| StableVITON[23] | 15.46 | 16.54 | 0 |
| **Tunnel Try-on** | **69.92** | **67.62** | **88.07** |
| ClothFormer\* [21] | 30.8 | 26.0 | 39.6 |
| **Tunnel Try-on\*** | **69.2** | **74.0** | **60.4** |

In Table 1, we conduct quantitative experiments with both image-based and video-based metrics. For image-based evaluation, we utilize structural similarity (SSIM) [41] and learned perceptual image patch similarity (LPIPS) [48]. These two metrics are used to evaluate the quality of single-image generation under the paired setting. The higher the SSIM and the lower the LPIPS, the greater the similarity between the generated image and the original image.

For video-based evaluation, we employ the Video Frechet Inception Distance (VFID) [8] to evaluate visual quality and temporal consistency. The FID [15] measures the diversity of generated samples. Furthermore, VFID employs 3D convolution to extract features in both temporal and spatial dimensions for better measures. Two

CNN backbone models, namely I3D [2] and 3D-ResNeXt101 [14], are adopted as feature extractors for VFID.

Table 1 demonstrates that on the VVT dataset, our Tunnel Try-on outperforms others in terms of SSIM, LPIPS, and VFID metrics, further confirming the superiority of our model in image visual quality (similarity and diversity) and temporal continuity compared to other methods. It's worth noting that we have a substantial advantage in LPIPS compared to other methods. Considering that LPIPS is more in line with human visual perception compared to SSIM, this highlights the superior visual quality of our approach.

Considering that the quantitative metrics could not perfectly align with the human preference for generation tasks, we conducted a user study to provide more comprehensive comparisons. We organized a group of 10 annotators to make comparisons on the 130 samples of VVT test set. We let different methods generate videos for the same input, and let the annotators pick the best one. The evaluation criteria included three aspects: quality, fidelity, and smoothness. "Quality" denotes the image quality, encompassing aspects like artifacts, noise levels, and distortion. "Fidelity" measures the ability to preserve details compared to the reference clothing image. "Smoothness" evaluates the temporal consistency of the generated videos. Note that ClothFormer is not open-sourced but it provides 25 generation results. We conduct an individual comparison in the bottom block of Table 1 for the 25 provided results between ClothFormer and our method. Results show that our method demonstrates significant superiority over the others.

## 4.4 Qualitative Analysis

Due to the limited diversity and simplicity of samples in the VVT dataset, it fails to represent the scenarios encountered in actual video try-on applications. Therefore, we provide additional qualitative results on our own dataset to highlight the robust try-on capabilities and practicality of Tunnel Try-on. Figure 1 illustrates various results generated by Tunnel Try-on, including scenarios such as changes in the size of individuals due to person-to-camera distance variation, the parallel motion relative to the camera, and alterations in background and perspective induced by camera angle changes. By integrating the focus tunnel strategy and focus tunnel enhancement, our method demonstrates the ability to effectively

(a) input      (b) w/o tunnel      (c) w/ tunnel

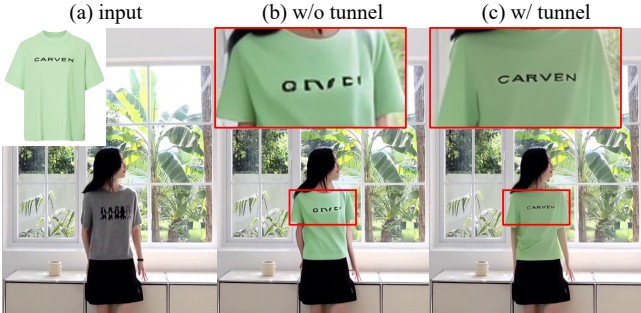

**Figure 5: Qualitative ablations for the focus tunnel. This zoom-in strategy brings notable improvements for preserving the fine details of the clothing.**

adapt to different types of human movements and camera variations, resulting in high-detail preservation and temporal consistency in the generated try-on videos.

Moreover, unlike previous video try-on methods limited to fitting tight-fitting tops, our model can perform try-on tasks for different types of tops and bottoms based on the user's choices. Figure 4 presents some try-on examples of different types of bottoms.

## 4.5 Ablation Study

We conducted ablation experiments for Tunnel Try-on to explore the impact of focus tunnel extraction (Section 3.3), focus tunnel enhancement (Section 3.4), and environment encoding (Section 3.5). We conduct both qualitative and quantitative ablations on our collected dataset to assess their performance.

In Table 3, we provide quantitative metrics related to the ablation experiments. The Focus Tunnel strategy significantly improves the model's SSIM and LPIPS metrics, but it leads to a certain degree of decrease in the VFID metric. This indicates that the Focus Tunnel can effectively enhance the quality of frame generation but may introduce more flickering, reducing the temporal consistency of the video. However, with the tunnel enhancement, the network's VFID shows a significant improvement, while the SSIM also increases. Lastly, although the environment encoder does not exert a significant impact on quantitative metrics, we observed that it contributes to the generation of the background environments around the clothing, as demonstrated in Figure 7. We conduct a detailed analysis of each component in the following paragraphs.

As shown in Figure 5, the impact of the Focus Tunnel Strategy is evident. Without the focus tunnel, there exists obvious distortion in the details of the logos. However, after zooming in on the tunnel regions with a close-up shot of the clothing. The detailed information of the garments could be significantly better preserved.

In Figure 6, we investigate the effectiveness of the tunnel enhancement. As depicted in the red box area, when the tunnel enhancement is not employed (first row), the clothing textures exhibit variations and flickering over time, leading to decreased temporal consistency in the generated video.

Figure 7 illustrates the impact of the environment encoder on the generation results. Since the environment encoder can extract overall context information outside the focus tunnel, it can enhance the quality of the background around the garment, making it more

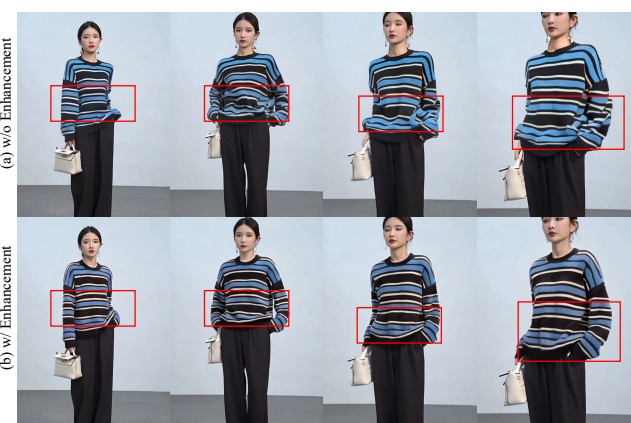

**Figure 6: Qualitative ablations for the tunnel enhancement. It assists in generating more stable and continuous textures.**

(a) w/o Env    (b) w/ Env    (c) w/o Env    (d) w/ Env

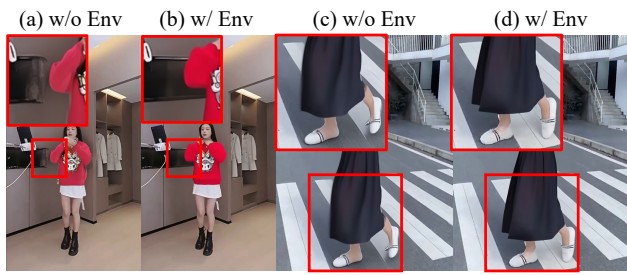

**Figure 7: Qualitative ablations for the environment encoder. The global context contributes to the recovery of the background around the clothing regions.**

**Table 3: Quantities ablations for the core components. "Tunnel", "Enhance", and "Env" denote the focus tunnel, the tunnel enhancement, and the environment encoder respectively.**

| Tunnel | Enhance | Env | SSIM↑ | LPIPS↓ | $VFID_{I3D}$↓ | $VFID_{ResNeXt}$↓ |
|--------|---------|-----|-------|--------|---------------|-------------------|
|        |         |     | 0.801 | 0.061  | 6.103 | 8.751 |
| ✓      |         |     | 0.877 | 0.052  | 6.759 | 9.034 |
| ✓      | ✓       |     | **0.914** | 0.049 | 5.997 | 8.356 |
| ✓      | ✓       | ✓   | 0.909 | **0.042** | **5.901** | **8.348** |

consistent with high-level semantic information about the environment. As shown in Figure 7, when the environment encoder is added, the generation errors in the textures of the walls and zebra crossings near the human are corrected.

## 5 Conclusion

We propose the first diffusion-based video virtual try-on model, Tunnel Try-on. It outperforms existing alternatives in both qualitative and quantitative comparisons. Leveraging the focus tunnel, tunnel enhancement, and environment encoding, it can adapt to diverse camera movements and human motions in videos. Trained on real datasets, our model could handle virtual try-on in videos with complex backgrounds and high-quality clothing textures. Serving as a practical tool for the fashion industry, Tunnel Try-on provides new insights for future research in virtual try-on applications.

## Acknowledgments

This work is supported by Hubei Provincial Natural Science Foundation of China No.2022CFA055 and the National Natural Science Foundation of China No.62176097.

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
