# OpenReview forum: "Tunnel Try-on: Excavating Spatial-temporal Tunnels for High-quality Virtual Try-on in Videos"
_acmmm.org/ACMMM/2024/Conference — MM2024 Poster_

### Official Review · Reviewer_8zFG · 2024-05-13

**Rating:** 2
**Confidence:** 3

**Summary:**

The goal of this work is to tackle video virtual try-on task. To achieve this purpose, this paper builds a new virtual try-on dataset, which consists of a total of 5,350 video-image pairs. In addition, this work constructs the focus tunnel to achieve better performance, and uses Kalman filter, tunnel position embedding and environment context to further enhance the authenticity of try-on results.

**Strengths:**

1. This work proposes the first diffusion-based video virtual try-on model and has better performance than previous work.
2. This work collects a new virtual try-on dataset which consists of a total of 5,350 video-image pairs.
3. This work constructs the focus tunnel to emphasize the clothing region in videos and utilizes Kalman filter, tunnel position embedding and environment context to further enhance model's performance.

**Limitations:**

1. The contribution of tunnel extraction is not novel enough. It zooms in the agnostic person but does not deal with the clothing. So it does not seem to be significantly helpful in enhancing the details of the original clothing. In addition, tunnel extraction may destroy the coherent motion in the original video.
2. It may be more reasonable to provide a detailed comparison between the dataset proposed by this work and the VVT dataset, as well as give some visual comparisons. What is the resolution of the proposed dataset?
3. The experiments in this work seem somewhat insufficient. The paper seems to only provide quantitative comparison experiments and ablation study on the VVT dataset. It might be better to add more experiments.
4. It might be better to explain the implementation details of the temporal aggregation technique to combine different video clips in the testing phase.
5. For the video virtual try-on task, is it possible to use the image-based virtual try-on method to generate a certain frame, and then use the video pose transfer method to generate the video try-on result? It might be better to conduct such experiments in contrast to the approach of this work.

**Suitability:**

3

---

### Official Review · Reviewer_pAJK · 2024-05-20

**Rating:** 5
**Confidence:** 3

**Summary:**

This paper propose a diffusion-based video virtual try-on model and design a novel technique of constructing the focus tunnel to emphasize the clothing region and generate coherent motion in videos. Equipped with these techniques, Tunnel Try-on keeps the fine details of the clothing and synthesizes stable and smooth videos. Demonstrating significant advancements, Tunnel Try-on could be regarded as the first attempt toward the commercial-level application of virtual try-on in videos.Experiments show that the proposed video virtual try-on method demonstrates state-of-the-art performance in complex scenarios.

**Strengths:**

The overall structure of the paper is good, the idea is efficient, and the proposed experimental results show the effectiveness of the proposed method.

**Limitations:**

The proposed algorithm in this paper includes some modules and techniques, but due to the incomplete ablation studies, it is difficult to clearly identify the contributions of the paper.，e.g.
1、Innovativeness: The paper claims to be the first to propose a video algorithm based on a diffusion model. However, from the perspective of the algorithm and network structure, it is still based on an image diffusion model and has not truly achieved a complete video diffusion model. Therefore, the novelty of this approach is questionable.
2、Necessity of Improvement: The third innovation point of the paper is the introduction of the Kalman filter. However, the paper only provides a simple formula (Formula 3) and its implementation related to the Kalman filter. The experimental section does not offer an in-depth analysis of the Kalman filter. Thus, the necessity of the Kalman filter in the algorithm presented in this paper is debatable.
3、Network Structure: From the perspective of network structure, this paper does not present substantial improvements. However, the experimental results show significant performance enhancements. Therefore, the authors need to provide relevant experiments to demonstrate whether the performance improvement is related to the use of a large model CLIP.

**Suitability:**

2

---

### Official Review · Reviewer_bhNs · 2024-05-23

**Rating:** 6
**Confidence:** 2

**Summary:**

The paper presents a new framework for video virtual try-on called "Tunnel Try-on". The main goal is to improve the quality of virtual try-on in videos, including complex natural scenes. The proposed method addresses challenges such as handling complex clothing and backgrounds, adapting to different types of human movements, etc.

**Strengths:**

1. A novel approach called "Tunnel Try-on" that leverages diffusion models for high-quality VVT in videos.
2. The development of a technique called the "focus tunnel" to emphasize the clothing region and generate coherent motion in videos.
3. Enhancing strategies like incorporating the Kalman filter to smooth the focus tunnel and injecting tunnel position embedding into the attention layers to improve the generation quality.
4. Sufficient experiments and supplementary materials show the effectiveness of the designed method.
5. Pretty good writing.

**Limitations:**

Wrong symbol syntax, i.e. the parameter of model in 3.1 (the 314th line).

**Suitability:**

3

---

### Meta-Review · Area_Chair_d8RN · 2024-07-05

**Recommendation:** Accept (Poster)
**Confidence:** 4

**Metareview:**

After considering all reviews, the rebuttal, and the subsequent discussion, AC decides to accept the paper.